# Decision Method of Optimal Needle Insertion Angle for Dorsal Hand Intravenous Robot

**DOI:** 10.3390/s23020848

**Published:** 2023-01-11

**Authors:** Zihan Zhu, Kefeng Li, Guangyuan Zhang, Hualei Jin, Zhenfang Zhu, Peng Wang

**Affiliations:** 1School of Information Science and Electrical Engineering, Shandong Jiaotong University, Jinan 250357, China; 2Institute of Automation, Shandong Academy of Sciences, Jinan 250013, China

**Keywords:** imaging system, linear structured light scanning, plane fitting, needle entry angle

## Abstract

In the context of COVID-19, the research on various aspects of the venipuncture robot field has become increasingly hot, but there has been little research on robotic needle insertion angles, primarily performed at a rough angle. This will increase the rate of puncture failure. Furthermore, there is sometimes significant pain due to the patients’ differences. This paper investigates the optimal needle entry angle decision for a dorsal hand intravenous injection robot. The dorsal plane of the hand was obtained by a linear structured light scan, which was used as a basis for calculating the needle entry angle. Simulation experiments were also designed to determine the optimal needle entry angle. Firstly, the linear structured optical system was calibrated and optimized, and the error function was constructed and solved iteratively by the optimization method to eliminate measurement error. Besides, the dorsal hand was scanned to obtain the spatial point clouds of the needle entry area, and the least squares method was used to fit it to obtain the dorsal hand plane. Then, the needle entry angle was calculated based on the needle entry area plane. Finally, the changes in the penetration force under different needle entry angles were analyzed to determine the optimal needle insertion angle. According to the experimental results, the average error of the optimized structured light plane position was about 0.1 mm, which meets the needs of the project, and a large angle should be properly selected for needle insertion during the intravenous injection.

## 1. Introduction

The COVID-19 outbreak has caused a cumulative total of more than 630 million infections worldwide; some of these infections are among medical personnel [1], and the primary cause of their infections is prolonged exposure to high-risk environments [2]. In contrast, as an essential medical treatment, dorsal hand intravenous injection often requires prolonged direct contact between medical personnel and patients. This vastly increases the risk of cross-infection [3,4], and the success rate of puncture depends largely on the accumulated experience of the medical staff [5,6,7]. Prolonged wearing of goggles, protective clothing, and other facilities can further increase the failure rate of puncture [8]. Suppose robots are used to replace medical staff to administer dorsal hand IVs to infected patients. In that case, they could effectively reduce the risk of infection to medical staff, significantly improve puncture success rates and efficiency, and significantly reduce the shortage of medical staff [9].

Current research in the field of intelligent injection robotics focuses on vein imaging [10,11], detection and segmentation [12,13], decision on needle insertion point location [14,15], and robot system design [16,17]. At the same time, fewer studies are related to the needle insertion angle for dorsal hand intravenous injection robots. Li et al. [18] investigated the alignment of the puncture needle and vessel orientation for venipuncture robots, experimenting with the effect of three aspects of needle entry: velocity, deflection angle, and pitch angle on the magnitude of puncture force, but did not indicate how the entry and exit angles were determined. Peng Qi’s team [19] at Tongji University proposed a method to determine the angle of elbow vein entry based on ultrasonic evaluation. This method takes the angle between the needle axis and the patient’s entire arm as the entry angle, selecting a large or small entry angle roughly by ultrasonic evaluation of vessel thickness. Nevertheless, it only targets the vein at the elbow, which is thicker, more prominent, and easier to identify than the dorsal hand. Moreover, due to individual differences, patients sometimes feel significant pain.

To solve the above problems, this paper investigates the optimal needle insertion angle decision for the dorsal hand intravenous injection robot. The needle insertion angle is the surface of the dorsal hand as a reference surface [20]. Since the needle entry area is only a small area, the dorsal hand can be obtained by replacing the curved surface with a flat one, which requires obtaining depth information on the dorsal hand. However, it is impossible to recover the spatial coordinates from the pixel coordinates due to the lack of constraints for individual cameras. To obtain the absolute spatial coordinates, the following methods are commonly used to add constraints: increasing the number of cameras, changing the camera position, and adding additional sensors (such as line structured light) [21].

Chen Lin et al. [22] designed a new precise measurement method based on binocular vision and proposed a novel stereoscopic matching algorithm combined with epipolar geometry and cross-ratio invariance (CMEC) to improve the accuracy of binocular vision stereoscopic matching. Yunpeng Ma et al. [23] developed an ROV (Remotely Operated Vehicle) based on a binocular vision system to carry out underwater crack detection in real-time. Pranjali Pathre et al. [24] proposed and showcased a monocular multi-view layout estimation for warehouse racks and shelves and used a single monocular camera to portray a 3D rendering of a warehouse scene in terms of its semantic components. Danila Rukhovich et al. [25] proposed ImVoxel-Net, a novel fully convolutional method of 3D object detection based on posed monocular or multi-view RGB images, which successfully handles both indoor and outdoor scenes. Jia Liang et al. [26] proposed a three-dimensional pavement detection system based on linear structured light, which significantly reduced costs without decreasing the test accuracy.

Comparing the three methods: increasing the number of cameras requires matching the positions of corresponding points in a multi-camera view, which is particularly sensitive to ambient illumination and has a high computational complexity [27]. The method of changing camera positions requires a real-time estimation of inter-camera poses and has a scale problem [28]. Increasing sensors requires additional equipment but can obtain depth information quickly and without scale problems [29], so this paper chooses the method of monocular cameras combined with linear structured light scanning.

In the previous research, the group completed the decision of needle insertion point location and obtained high-precision needle entry point coordinates and entry direction [30], while this paper focuses on the optimal entry angle decision. The spatial point clouds are obtained by scanning the dorsal hand with a linear structured light, and the dorsal surface of the hand is obtained by fitting the spatial point clouds of the needle entry area. To address the problem of inaccurate measurement results due to errors in the fitting process, the error function has been constructed and solved iteratively by the optimization method to eliminate the errors. After the optimized dorsal surface of the needle entry area is obtained, the angle between the direction vector of the needle axis and the dorsal surface of the hand is the needle entry angle. The simulation experiments are designed to investigate the changes in the puncture force under different needle entry angles, and the optimal needle entry angle decision is completed.

## 2. Implementation Method

The research group researched non-contact dorsal hand intravenous injection and designed an intelligent robot for dorsal hand intravenous injection, and the overall workflow is shown in Figure 1.

Through the previous work, our group has completed the recognition segmentation of dorsal hand vein images. The pixel coordinates of the dorsal hand vein are extracted by using the method of needle insertion position decision based on the pruning algorithm (PT Running). Considering the sectional area and bending value of each vein in the needle insertion area, the optimal needle insertion point and direction of the dorsal hand vein are comprehensively determined [30]. In this paper, we focus on the decision of the optimal needle entry angle.

The needle entry angle of the dorsal hand is calculated based on the plane of the dorsal hand needle entry area. Therefore, the depth information of the dorsal hand needs to be obtained first. Since the camera acquires a two-dimensional image, it is necessary to build a camera imaging model in order to determine the relationship between the spatial point clouds on the dorsal hand and their corresponding points in the image. The internal and external parameters and distortion coefficients of the camera are obtained through camera calibration. However, a single camera cannot recover from two-dimensional coordinates to three-dimensional coordinates, so this paper introduces linear structured light to assist in obtaining depth information of the dorsal hand.

Firstly, the dorsal hand is scanned using linear structured light, and the scanned images are acquired isometrically. Then, the images are processed to extract the centerline of the structured light strip, and the spatial point clouds of the dorsal hand are obtained and optimized to eliminate the error. The spatial point clouds of the entry and exit needle area are screened and fitted by the least squares method [31] to finally obtain the plane of the entry needle area. The whole workflow is shown in Figure 2.

The experimental platform model designed in this paper is shown in Figure 3. The camera and motor-axis platform are fixed together so that the camera’s optical axis passes through the dorsal hand. The structured light projector is fixed on the slide table on the axis so that the structured light plane is scanned perpendicular to the direction of the axis movement.

### 2.1. Camera Calibration

Camera calibration aims to obtain the camera’s internal and external parameters and aberration coefficients to facilitate accurate subsequent calculations. Currently, the camera calibration process has been developed more maturely, and various calibration methods exist. This paper uses the calibration method of Zhang; this method is a simple and flexible operation, and its accuracy can meet the project requirements [32]. In this paper, we chose the 8 × 6 size checkerboard plane calibration board, as shown in Figure 4.

Because Zhang’s calibration method uses different positions between the camera and calibration board to solve for the parameters, it is necessary to acquire calibration board images at different distances and angles. Considering the projection relationship between the natural world and the image, suppose a spatial point exists P = (Xc,Yc,Zc) with the corresponding pixel coordinates P = (u,v). The transformation relationship between P and p [11] can be expressed as:(1)Zc[uv1]=[fx  0  uo  0 0  fy  vo  0 0   0   1   0][XcYcZc1]=[K  03×1][XcYcZc1]=M[XcYcZc1],
where K∈ℝ3×3 is the in-camera parameter matrix; and M∈ℝ3×4 is the transformation matrix. Since M is not invertible, the spatial coordinates cannot be obtained from the pixel coordinates.

Due to the camera production process, the image captured by the camera will have some degree of distortion, called camera aberration. The general camera aberration model [16] can be expressed as:(2)$$\begin{array}{r} x_{corrected}=\;x\left(1+k_{1}r^{2}+k_{2}r^{4}+k_{3}r^{6}\right)+[2p_{1}xy+p_{2}\left(r^{2}+2x^{2}\right)],\\ y_{corrected}=\;y\left(1+k_{1}r^{2}+k_{2}r^{4}+k_{3}r^{6}\right)+\left[p_{1}\left(r^{2}+2y^{2}\right)+2p_{2}xy\right]\\ \left(r^{2}=x^{2}+y^{2}\right) \end{array}$$
where (x,y) on the right side of the equation is the ideal point coordinate in the obtained image, but with aberrations, $\left(x_{corrected},y_{corrected}\right)$ on the left side of the equation is the actual point coordinate after aberration correction, $k_{1}$, $k_{2}$, and $k_{3}$ are the radial aberration coefficients, and $p_{1}$ and $p_{2}$ are the tangential aberration coefficients.

### 2.2. Structured Light Plane Calibration

Due to a single camera’s conditions, it is impossible to recover the spatial coordinates from the pixel coordinates. To obtain the spatial coordinates of the dorsal hand into the needle region, this paper uses linear structured light to assist the measurement to obtain the depth information, as follows.

From Equation (1), if the image coordinates (u,v) and $Z_{c}$ are known, the camera coordinates $\left(X_{c},Y_{c},Z_{c}\right)$ can be found as follows:(3)$$\begin{array}{l} X_{c}=\frac{u-u_{0}}{f_{x}}Z_{c}\\ Y_{c}=\frac{v-v_{0}}{f_{y}}Z_{c}, \end{array}$$

To obtain $Z_{c}$, let the equation of the light plane of the line structure under the camera coordinate system be:(4)$$AX_{c}+BY_{c}+CZ_{c}+D=0,$$

Substituting Equation (3) into Equation (4) yields:(5)$$Z_{c}=-\frac{D}{\frac{A\left(u-u_{0}\right)}{f_{x}}+\frac{B\left(v-v_{0}\right)}{f_{y}}+C}$$

Therefore, to obtain $Z_{c}$, it is necessary to first determine the plane equation through the linear structured light plane calibration. Because the linear structured light projector needs to move along a fixed direction, the structured light plane equation at any position needs to be determined. So, a multi-structured light plane calibration is required in the next step. The structured light projector is moved at the same distance interval along the axis. Combined with the calibration method of single structured light plane, *n* sets of structured light strip point coordinates and *n* structured light planes can be obtained. In turn, the public normal vector of the structured light planes and the equation of the structured light planes at any position are obtained.

However, the camera coordinates of the points on the structured light strips obtained above do not consider the effects of camera distortion and other constraints, and the calculation results will deviate from the actual positions. In this paper, we take into account the reprojection, linear, and plane constraints in the structured light plane calibration process and use them to construct an error function, which is solved iteratively by the optimization method [33] to eliminate the error.

#### 2.2.1. Single Structured Light Plane Calibration

The principle of the single structured light plane calibration is shown in Figure 5, where K is the intra-camera parameter matrix, dist is the aberration coefficient, and A, B, C, D are the structured light plane equation parameters corresponding to Equation (4).

The specific algorithm flow is shown in Algorithm 1. The inputs are multiple sets of calibration plate images IMG and the world coordinates CW of the corner points. The outputs are the parameters A, B, C, D of the structured light plane equation. First, the calibration plate image IMG is traversed, and the corner pixel coordinates are obtained using the cv2.findChessboardCorners() method. Then, according to the corresponding relationship between corner pixel coordinates and world coordinates, the rotation matrix R, translation matrix T, and homography matrix H of the image are obtained using the cv2.solvePnP() method and cv2.findHomography() method, respectively. The cv2.perspectiveTransform() method is used to perform the perspective transformation of corner pixel coordinates and cut out the transformed calibration board ROI area. Then, the function extractLaser() is designed to extract the ROI area structured light strip and the pixel coordinates of the structured light strip, before transformation is obtained through the inverse perspective transformation using the cv2.perspectiveTransform() method. Then, the camera coordinates of the structured light strip are calculated by rotation matrix R and translation matrix T. After obtaining the camera coordinates of the structured light strip for each group of images, the optimization method optimizeLaser() is designed to optimize them. Finally, the parameters A, B, C, D of the structured light plane equation are obtained by fitting the camera coordinates of the optimized structured light strip with the np.linalg.svd() method.
**Algorithm 1:** Calibration of Structured Light Plane**Input:** calibration board image IMG; corner world coordinates CW**Output:** parameters of structured light plane A, B, C, D
1.  **for** i **in** IMG **do**
2.    Begin 
3.    // Extract pixel coordinates of corner points of calibration board 
4.    CP = cv2.findChessboardCorners(i) 
5.    // Calculate the rotation matrix R, translation matrix T and homography matrix H 
6.    R, T = cv2.solvePnP(CW, CP) 
7.    H = cv2.findHomography(CP, CW) 
8.    // Perspective transformation 
9.    CT = cv2.perspectiveTransform(CP, H) 
10.   ROI = cv2.warpPerspective(IMG, H, CT) 
11.   // Extract ROI area structure light strip 
12.   LT = extractLaser(ROI) 
13.   // Inverse perspective transform 
14.   LP = cv2.perspectiveTransform (LT, H^-1^) 
15.   // Calculate the camera coordinates of the structured light strip 
16.   LC = R @ LP + T 
17.   End
18. // Optimize
19. LO = optimizeLaser(LC)
20. // Structured light plane fitting
21. A, B, C, D = np.linalg.svd(LO)

The following describes the specific process of single structured light plane calibration.
The world coordinate system is established by taking the upper left corner of the calibration board plane as the origin, the long side along the checkerboard grid as the x-axis and the short side as the y-axis, and the perpendicular to the calibration board outward as the z-axis to obtain the corner point world coordinates. Multiple sets of calibration board images are collected with and without structured light in the same position, as shown in Figure 6. Then processing is carried out to obtain the corner point pixel coordinates.The rotation matrix R and translation matrix T for each group of images is calculated, and the homography matrix H is obtained according to the correspondence between the pixel coordinates of the corner points and their world coordinates. After graying the image, the corner area is transformed to the upper left through perspective transformation. The corner ROI area of the calibration board is intercepted according to the transformed corner coordinates, as shown in Figure 7.Each grid in the calibration board ROI area is divided according to the corner coordinates. The threshold value is set to preliminarily screen out the grids with a pixel difference more significant than 30, and binary processing is carried out. Then, the contour with the most significant area is fitted to obtain the smallest outer rectangle, and the contours with a ratio greater than 0.25 are eliminated according to the rectangle height-to-width ratio. Finally, the slope between two grids after secondary screening is calculated, the grids with apparent deviations are eliminated, and the final result is shown in Figure 8.Further refinement of the obtained contour is carried out, but the refinement of the contour will have the phenomenon of “warping.” So, the coordinates of the contour points are sorted and then sliced, and the head and tail points are removed and refined. The result is shown in Figure 9.The centerline of the structured light strip is obtained by fitting the optimized contour point coordinates. The corresponding pixel coordinates are obtained by inverse perspective transformation, as shown in Figure 10.The pixel coordinates of the extracted structured light strip quadrature points on each calibration plate image are denoted as $P_{image}$. Each image’s rotation matrix R and translation matrix T is used to calculate their corresponding camera coordinates $P_{camera}$ through the Formula (6). As shown in Figure 11, the calculated camera coordinates of each set of structured light strip quadrature points are displayed. It can be seen that all points are nearly on the same plane, i.e., the structured light plane, but there are still some errors.
(6)$$P_{camera}=RP_{image}+T.$$The calculated camera coordinates of the structured light strip quadrature points are optimized to eliminate the effect of errors. Then, the structured light plane equations are obtained by fitting the optimized structured light strip camera coordinates to the points.

Next, we introduce the error optimization method in detail.

There is always a difference between the 2D image coordinates calculated from the 3D spatial coordinates according to the inverse perspective transformation and the actual image coordinates, that is, the reprojection error. Because the structured light strip is a straight line on the plane calibration plate, the centerline coordinates of the structured light strip are not in a straight line after transforming to the space point coordinates, and there is a linear error. Moreover, because all structured light strips are issued by the same location structured light projector, the structured light strip space points should logically lie in the same plane. However, the calculated three-dimensional space points will not be accurately distributed on a plane, and there is a plane error. Thus, in this paper, the error function is constructed by considering the above three constraints with the aid of a calibration board.
(7)$$Loss=ln\left({\left\|u_{o}-u_{c}\right\|}_{2}^{2}\right)+{\left(1-\left|cos\theta \right|\right)}^{2}+{\left[ln(1+d)+ln({\left\|n\right\|}_{2}^{2})\right]}^{2}.$$

The error function consists of three parts: the first part $ln\left({\left\|u_{o}-u_{c}\right\|}_{2}^{2}\right)$ is a reprojection constraint on the 3D spatial point coordinates so that the Euclidean distance between the original pixel coordinate point $u_{o}$ and the pixel coordinate point $u_{c}$ calculated by Equation (1) is minimized. The optimization result is shown in Table 1. “Not Optimized” is the error between the pixel coordinate points before the optimization and the initial pixel coordinate points, and “Optimized” is the error between the pixel coordinate points after the optimization and the initial pixel coordinate points.

The second part ${\left(1-\left|\cos\theta \right|\right)}^{2}$ is a linear constraint on the structured light strip camera coordinates because the calibration plate is flat, so the structured light strip point coordinates should be on a straight line after the conversion to spatial point coordinates. This paper constrains the angle between three adjacent points in each structured light strip spatial point to be as small as possible, as shown in Figure 12, so that the angle *θ* between three adjacent points tends to be 0° or 180°.
(8)$$\cos\theta =\cos<\vec{ab},\;\vec{ac}>.$$

The third part ${\left(ln(1+d)+ln({\left\|n\right\|}_{2}^{2})\right)}^{2}$ (where $n={\left[A\;B\;C\right]}^{T}$) is a plane constraint on the 3D spatial point coordinates. Because the same structured light projector generates all structured light strips, ideally, the collected 3D spatial points should be on the same plane. For this reason, this paper makes the distance between the spatial points and the structured light plane as small as possible, as shown in Figure 13, so that d tends to 0. The purpose of introducing $ln({\left\|n\right\|}_{2}^{2})$ is to ensure that the modulus length of the plane normal vector is 1.

The logarithmic operation is introduced in the first and third terms of the error function, which aims to ensure that all three terms are in the same order of magnitude to avoid numerical problems. Finally, an iterative optimization algorithm is used to minimize the above errors to obtain the optimized camera coordinate points, and then the optimized camera coordinate points are fitted by the least squares method to obtain the structured light plane.

Figure 14 compares the camera coordinates of all structured light strip quadrature points before and after optimization. “Not Optimized” is the average distance from all points to the fitting plane before optimization, and “Optimized” is the average distance from all points to the fitting plane after optimization. As can be seen from the data in the figure, the camera coordinates of the optimized quadrature points are closer to the fitting plane.

#### 2.2.2. Multi-Structured Optical Plane Calibration

Since the structured light projector moves at a fixed distance along the axis, ideally, all structured light planes have a common plane normal vector (A,B,C). $D_{n}$ in the equation of the structured light plane is linearly related to the distance $x_{n}$ at which the structured light projector moves along the axis, which can be expressed as:(9)$$D_{n}=ax_{n}+b.$$

The structured light plane equation at any position can be obtained by using the least squares solution to obtain the parameters a and b. The principle of the multi-structured light plane calibration is shown in Figure 15. The parameters in the figure are basically the same as those in Figure 5, but the difference is that the structured light plane equation in multiple locations is required. After the single structured light plane calibration, the accurate camera coordinates of the 3D spatial points on the structured light plane can be obtained, while the multi-structured light plane calibration can obtain the structured light plane equation at any position by Equation (11). Combined with the single-structure optical plane calibration results, the exact camera coordinates of any spatial point can be obtained.

The specific process of the multi-structured optical plane calibration is described below.

1.Fix the initial positions of the linear structured light projector and the camera. Apply the single structured light plane calibration method to obtain the image coordinates, camera coordinates, and structured light plane equations of the points on the centerline of the structured light strip at the current position.2.Move the linear structured light projector 5 mm in the direction of the slide and apply the single structured light plane calibration method for this position.3.Repeat the above process to obtain a total of *n* sets of coordinates of points on the centerline of the structured light strip and the structured light plane equation. In this paper, *n* is taken as 13, and a total of 60 mm is moved along the axis.4.Optimize the 13 sets of structured light planes, so each structured light plane has a common plane normal vector, as shown in Figure 16, and the fitted structured light planes are parallel.

To verify the accuracy of the fitted structured light planes, the distance between every two planes is calculated by Equation (10). Let the equations of the two planes be $AX_{c}+BY_{c}+CZ_{c}+D_{i}=0$ and $AX_{c}+BY_{c}+CZ_{c}+D_{i+1}=0$, respectively, and then we can obtain:(10)$$d=\frac{\left|D_{i+1}-D_{i}\right|}{\sqrt{A^{2}+B^{2}+C^{2}.}}$$

The calculated results are shown in Table 2, and the average error is about 0.1 mm, which can prove that the results are relatively accurate.

The final purpose of the structured light plane calibration is to obtain the equation of the structured light plane at any position, as shown in Table 3 for the optimized structured light plane common normal vector (A,B,C) and the parameter vector (a,b) for solving D.

To verify the accuracy of the structured light plane calibration, this paper establishes a prediction model to compare the predicted values of the structured light plane equation parameter D with the actual values. The comparison results are shown in Table 4, and the average value of the difference is 0.174.

### 2.3. Optimization of Spatial Points during the Measurement

The optimization method for spatial points during the measurement is the same as the previous paper. The only difference is that the calibration plate used during calibration is a regular plane, while the dorsal hand is irregular. Therefore, linear constraints cannot be placed on the camera coordinate points. Therefore, the error function of the measurement is modified based on Equation (7), as shown in Equation (11).
(11)$$Loss=ln\left({\left\|u_{o}-u_{c}\right\|}_{2}^{2}\right)+{\left[ln(1+d)+ln({\left\|n\right\|}_{2}^{2})\right]}^{2}.$$

From Equation (11), it can be seen that the error function at the time of measurement is optimized only for reprojection error and plane error. The same iterative optimization algorithm described above eventually yields optimized and accurate camera coordinate points.

### 2.4. Fitting the Dorsal Hand Plane in the Needle Entry Area and Calculating the Needle Entry Angle

The image coordinates of the needle entry point with high accuracy and the coordinates of some pixels on the venous vessels, the direction of needle entry, have been obtained by the previous work [30]. After scanning the entire dorsal hand point cloud information, it is possible to perform a planar fit of the needle entry region.

Firstly, the image coordinate of the dorsal hand entry point is used as the center point to intercept the needle entry region. Then, the centerline of the structured light strip within the entry region is extracted separately. Then, the above optimization method is used to optimize, and the camera coordinate points in the dorsal hand needle entry region are obtained. Finally, the plane of the dorsal hand entry region is obtained by least squares fitting, the specific process of which is as follows.

Let the general expression of the plane equation be:(12)$$A^{\prime }x+B^{\prime }y+C^{\prime }z+D^{\prime }=0\left(C^{\prime }\neq 0\right).$$

Transform it into the following form:(13)$$z=-\frac{A^{\prime }}{C^{\prime }}x-\frac{B^{\prime }}{C^{\prime }}y-\frac{D^{\prime }}{C^{\prime }}.$$

Let $a=-\frac{A^{\prime }}{C^{\prime }}$, $b=-\frac{B^{\prime }}{C^{\prime }}$, $c=-\frac{D^{\prime }}{C^{\prime }}$. Then, Equation (13) can be expressed as:(14)z=ax+by+c.

At this point, its corresponding least squares matrix form is:(15)$$\left[\begin{array}{l} x_{1}\;y_{1}\;1\\ x_{2}\;y_{2}\;1\\ \;\;\;\;\cdot \cdot \cdot \\ x_{n}\;y_{n}\;1 \end{array}\right]\left[\begin{array}{l} a\\ b\\ c \end{array}\right]=\left[\begin{array}{l} z_{1}\\ z_{2}\\ \cdot \cdot \cdot \\ z_{n} \end{array}\right]\left(n\geq 3\right),$$
where $\left(x_{1},y_{1},z_{1}\right)$, $\left(x_{2},y_{2},z_{2}\right)$, …, $\left(x_{n},y_{n},z_{n}\right)$ is the dorsal hand space point clouds’ camera coordinates. By solving Equation (15) we obtain (a,b,c), that is, the plane equation of the dorsal hand needle entry area, corresponding to equation (14). This is used as a basis for calculating the puncture angle.

From Equation (12), the normal vector of the plane is:(16)$$\vec{n}=(A\prime ,B\prime ,C\prime ).$$

Let the camera coordinates of the entry point be $(x_{0},y_{0},z_{0}).$, and the direction vector of the puncture needle axis be:(17)$$\vec{s}=(m,n,p).$$

Then, the linear equation of the puncture needle axis is:(18)$$\frac{x-x_{0}}{m}=\frac{y-y_{0}}{n}=\frac{z-z_{0}}{p}.$$

The angle between the dorsal hand into the needle area plane and the puncture needle axis is noted as α, and the angle between the plane normal vector and the needle axis direction vector is noted as θ. The angle between the dorsal hand into the needle area plane and the puncture needle axis straight line is calculated as follows:(19)$$\begin{array}{l} \sin\alpha =\left|\cos\theta \right|=\left|\cos<\vec{n},\vec{s}>\right|=\left|\frac{\vec{n}\cdot \vec{s}}{\left|\vec{n}\right|\times \left|\vec{s}\right|}\right|\\ \;\;\;\;\alpha =\arcsin\left|\frac{\vec{n}\cdot \vec{s}}{\left|\vec{n}\right|\times \left|\vec{s}\right|}\right| \end{array}$$

### 2.5. Optimal Needle Angle Decision for Dorsal Hand Veins

During venipuncture of the dorsum of the hand, the puncture needle first pierces the skin surface of the dorsum of the hand into the subcutaneous fat. Then, it penetrates the venous vessel wall into the vessel, and finally pushes the needle in the direction of the vessel for about 5 mm with pressure. According to the previous investigation, it was found that the axial puncture force of the puncture needle changed obviously during the puncture process. With the puncture process, the puncture force gradually increases as a whole, and there will be a significant decrease in mutation at the moment of penetrating the vascular wall [34]. Therefore, the puncture force can be detected according to this characteristic, which can be used as an essential basis to judge whether it penetrates the vein.

Based on the existing conditions, this paper chooses to conduct venipuncture experiments by simulation. Venipuncture can be regarded as tissue penetration in a short period, which involves multiple interactions between various tissues. The LS-DYNA display kinetics module in ANSYS can perform drop, impact, and explosion experiments and analyze their processes. Therefore, this paper uses it to investigate the change curve of the puncture force under different entry angles and analyze it to determine the optimal entry angle.

## 3. Experimental Verifications

### 3.1. Experimental Platform Construction

According to the experimental model and demand analysis to carry out the equipment selection and build. This includes the camera, Sliding table, line structured light projector and hand model. The built experimental platform is shown in Figure 17.

### 3.2. Scanning to Obtain Spatial Point Clouds of the Dorsal Hand

According to the calibration method of the structured light plane proposed in this paper, the equation of the structured light plane at any position can be calculated. Next, scan the dorsal hand and extract the center line of the structured light strip of the hand back scanning image. Combined with the structured light plane equation under the corresponding position, the spatial point clouds on the dorsal hand are obtained. The specific experimental steps are as follows.
To make the scanning results more accurate, the structured light projector is first moved at 2 mm intervals, and the scanned image of the back of the hand was acquired, as shown in Figure 18.After graying the collected dorsal hand scanning image, the ROI area of the hand back is truncated. Then, the contrast of the ROI region is adjusted by gamma transformation to make the structured light strip region more prominent. The result is shown in Figure 19.Binarization of the gamma-transformed image and closed-operation processing of the image is carried out to ensure the integrity of the structured light strips. The result is shown in Figure 20.Refinement of the closed-operation processed image is carried out to extract the centerline of the structured light strip on the dorsal hand and save the point image coordinates. The results are shown in Figure 21.Based on the already obtained image coordinates of the dorsal hand entry point, the image coordinates of the dorsal hand point clouds in the area around the entry point are saved separately, as shown in Figure 22.

### 3.3. Fit the Dorsal Hand Plane in the Needle Entry Area and Calculate the Needle Entry Angle

After extracting the image coordinates of the structured light strip at the dorsal hand and needle insertion area, using the above optimization method, the world coordinates of the optimized structured light strip are obtained. Then, the plane equation of the entry region on the dorsal hand is obtained by fitting the plane to the entry region using the least squares method.
(20)−0.08x−0.53y−0.84z+373.38=0.

Based on the fitted equation of the dorsal hand entry area plane and the camera coordinates at the entry point, the entry angle can be calculated by Equation (19). Then, the plane of the longitudinal section of the vein, which is perpendicular to the plane of the hand dorsal needle insertion area, is taken as the auxiliary calculation plane. According to the needle insertion angle, the only needle insertion path pointing to the needle insertion point on the plane of the longitudinal section of the vein can be determined. The results are shown in Figure 23.

### 3.4. Optimal Needle Entry Angle Decision

Almost all biological tissues can be considered viscoelastic bodies, only differing in the degree of viscosity and elasticity. However, its mechanical properties are very complex and difficult to be fully demonstrated in a simulation environment, so, in engineering, most of the biological tissues are simulated by hyperelastic materials [35]. The common hyperelastic principal structure models are the following: Mooney Rivlin model, Yeoh model, Ogden model, Valanis Landel model, and Neo Hookean model. By reviewing the data and studying the mechanical properties of biological tissues such as skin, subcutaneous fat, and venous vessels [36], this paper adopts the Yeoh model for skin and venous vessels and the Mooney–Rivlin model for subcutaneous fat and muscle.

Firstly, the dorsal hand vein puncture model was established in SolidWorks in layers according to the actual scale and then imported into ANSYS. As shown in Figure 24, this mainly includes five parts: the puncture needle, the skin layer, the subcutaneous fat layer, the vein vessels, and the muscle layer. The material properties of each part are configured by reviewing relevant information to meet the experimental simulation requirements [37,38] including material density, material model and parameters, and failure properties. The mechanical property parameters of each layer organization are shown in Table 5.

The LS-DYNA module was used to perform venipuncture simulation experiments. Because the puncture model consists of five parts, the puncture process involves the setting of contact pairs between each part. A binding type of contact was used between the four layers of biological tissues, and a friction type of contact was used between the puncture needle and each biological tissue, with a friction coefficient set to 0.7. The “tetrahedral” method was introduced to divide the mesh to refine the biological tissues on the puncture path locally to improve the accuracy of the simulation. The puncture needle was set to move rigidly along the needle axis at a speed of 5 mm/s. A pressure of 0.7 kPa was applied to the inner wall of the venous vessel model to simulate venous blood pressure. To collect the puncture force information, nodal contact force probes were applied to the contact pairs between the puncture needle and each tissue separately.

During the puncture process, the node contact force collected by the three contact pairs is the force between the needle and the three layers of tissue. The superposition of the three forces is the puncture force information in the whole puncture process. The change curve of the overall puncture force is shown in Figure 25. The horizontal axis represents the time of puncture, and the vertical axis represents the size of the puncture force.

The changes in puncture force between the puncture needle and the skin layer, subcutaneous fat layer, and venous vessels are shown in Figure 26a, Figure 26b, and Figure 26c, respectively. It can be seen that at 0 s, the needle has first contact with the skin surface. At 0.07 s, the force between the needle and the skin layer decreases rapidly, indicating that it has penetrated the skin layer, corresponding to Figure 26a. At 0.14 s, the needle reached the subcutaneous fat layer. At 0.24 s, the force between the needle head and this layer decreased rapidly, indicating that it penetrated the subcutaneous fat layer, corresponding to Figure 26b. At 0.28 s, the needle has contact with the outer wall of the vein. It penetrates the blood vessel at 0.35 s, corresponding to Figure 26c. It can be observed that, in the puncture process of each tissue layer, the puncture force presents a trend of “increase decrease increase”, and when the lower epidermis is punctured, the puncture force is significantly reduced.

To study the influence of different needle entry angles on puncture force, in this paper, we designed puncture experiments at 30°, 45°, and 60°, and the results correspond to Figure 27a, Figure 27b, and Figure 27c, respectively. As can be seen from the figure, the smaller the angle of needle insertion, the longer the time required for the puncture process and the longer the distance of the puncture, at a constant speed. The overall trend of puncture force decreases as the angle of needle insertion increases. Therefore, it is medically recommended to use a larger entry angle to reduce the patient’s pain [39].

When designing the dorsal hand intravenous injection robot, the painless puncture method should also be used for reference in the puncture process, and the needle insertion angle should be appropriately increased. In addition, the robot is more accurate in controlling the large angle needle insertion operation compared with medical staff.

## 4. Conclusions

Since the outbreak of COVID-19, the demand for non-contact dorsal hand intravenous injection has continued to increase in medical institutions worldwide. This paper addresses the critical problem of the non-contact dorsal hand intravenous injection intelligent robot, aiming to solve the problem of optimal needle entry angle decision for the robot and ensure that patients can complete accurate needle entry with minimal pain. A monocular camera combined with a linear structured light scanning method was used to obtain spatial point clouds of the dorsal hand needle entry area for plane fitting. Additionally, a linear structured light system calibration and optimization method was proposed to eliminate measurement errors. Firstly, the linear structured light system was calibrated and optimized to obtain the equation of the structured light plane at any position. Then, the dorsum of the hand was scanned to obtain the spatial point cloud of the needle entry area and optimized to fit the dorsum of the hand plane. This plane was then used as the basis for calculating the needle entry angle, and experiments were designed to verify the feasibility of the method. Finally, simulation experiments were designed to analyze the changes in the penetration force at different needle entry angles and determine the optimal needle entry angle. However, this paper has only completed the optimal needle entry angle decision. The next step is to study how to ensure the accurate entry of the needle into the blood vessel after the needle is inserted and to cooperate with other functions to finally realize the non-contact dorsal hand intravenous injection intelligent robot design.

## Figures and Tables

**Figure 1 sensors-23-00848-f001:**
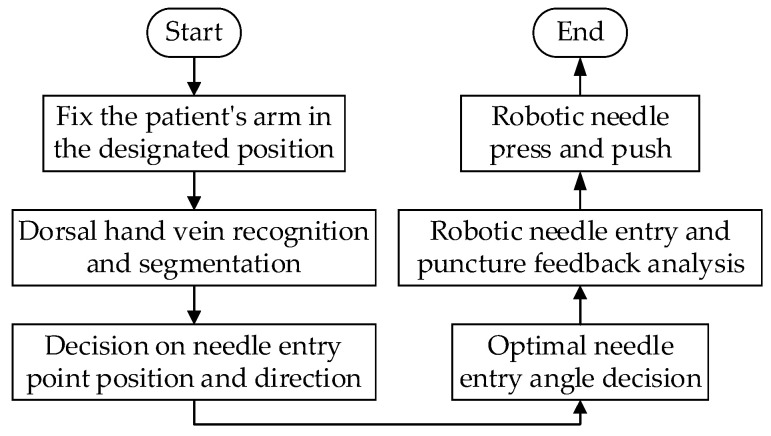
Dorsal hand intravenous injection intelligent robot workflow.

**Figure 2 sensors-23-00848-f002:**
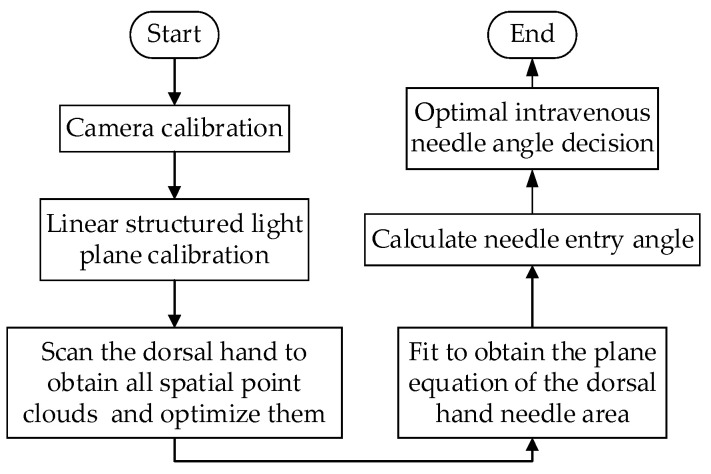
Workflow diagram.

**Figure 3 sensors-23-00848-f003:**
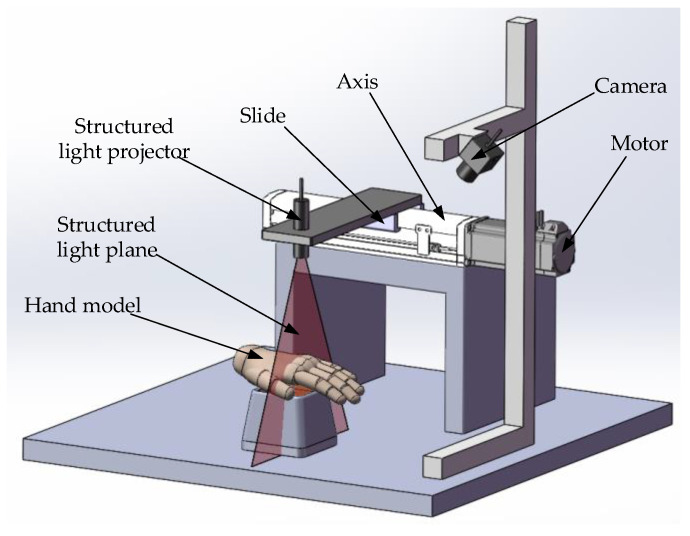
Workflow diagram.

**Figure 4 sensors-23-00848-f004:**
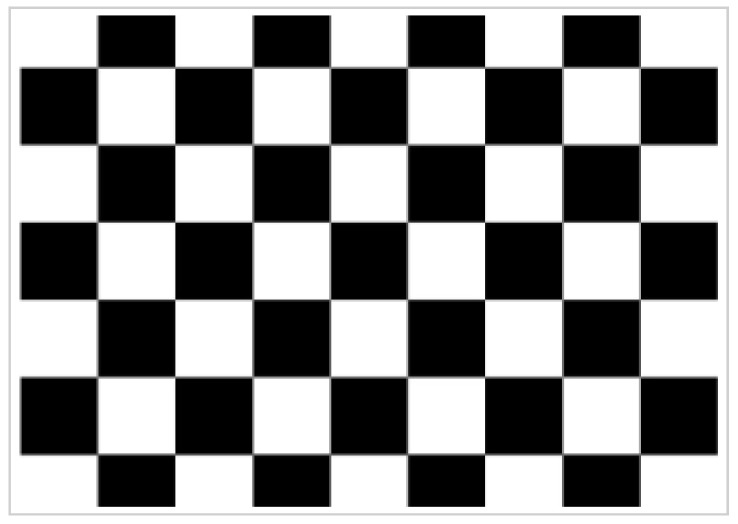
Checkerboard grid calibration pattern.

**Figure 5 sensors-23-00848-f005:**
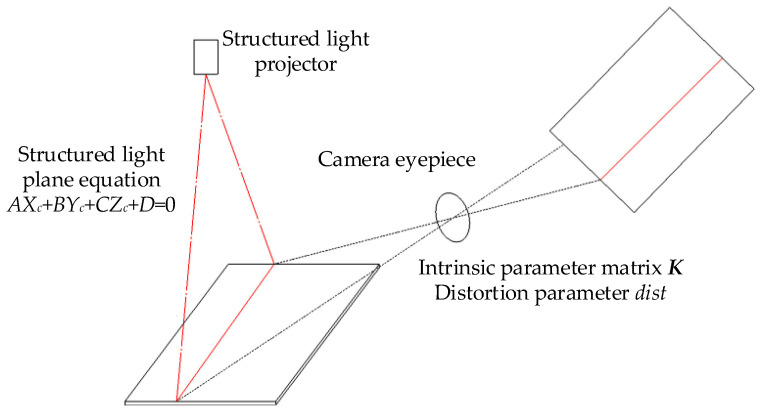
Diagram of calibration principle of the single structured light plane.

**Figure 6 sensors-23-00848-f006:**
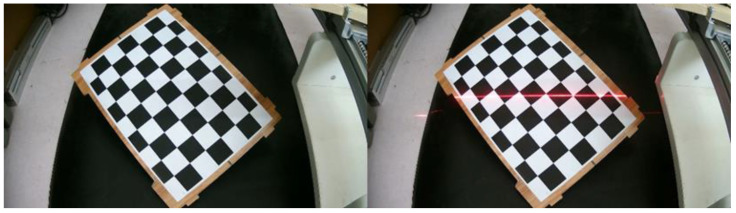
Image with and without structured light strip in the same position.

**Figure 7 sensors-23-00848-f007:**
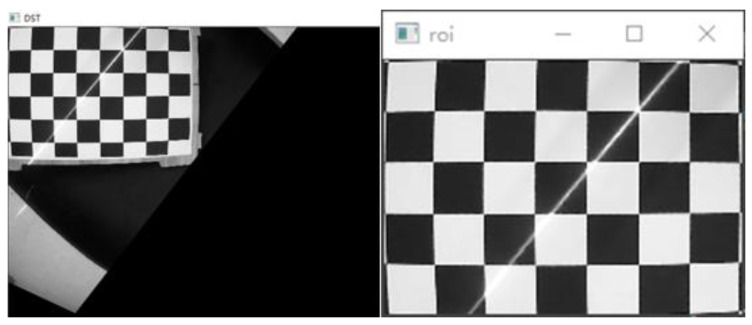
Intercepting the calibration plate ROI area.

**Figure 8 sensors-23-00848-f008:**
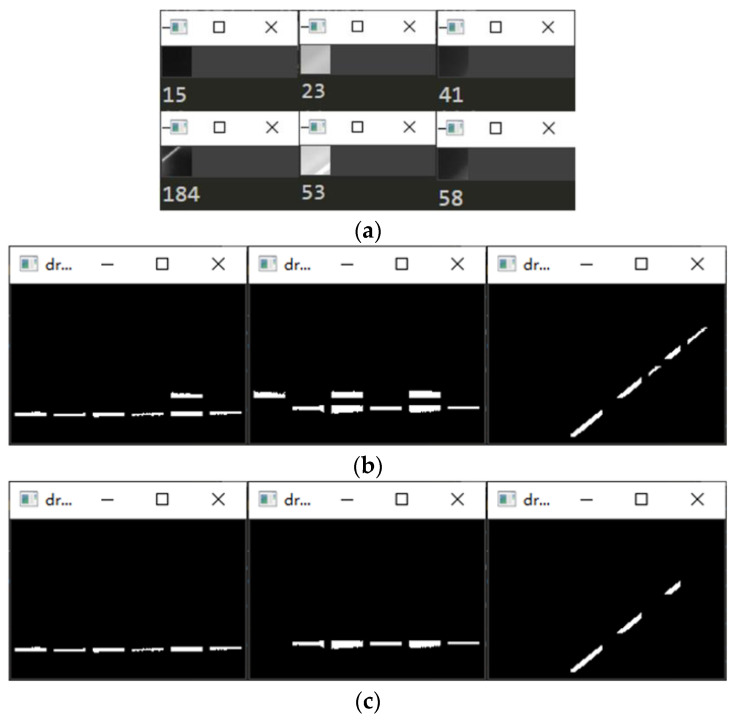
Structured light strip contour extraction results. (**a**) Pixel difference preliminary screening results; (**b**) Binarized image contour screening results; (**c**) Grids slope screening results.

**Figure 9 sensors-23-00848-f009:**
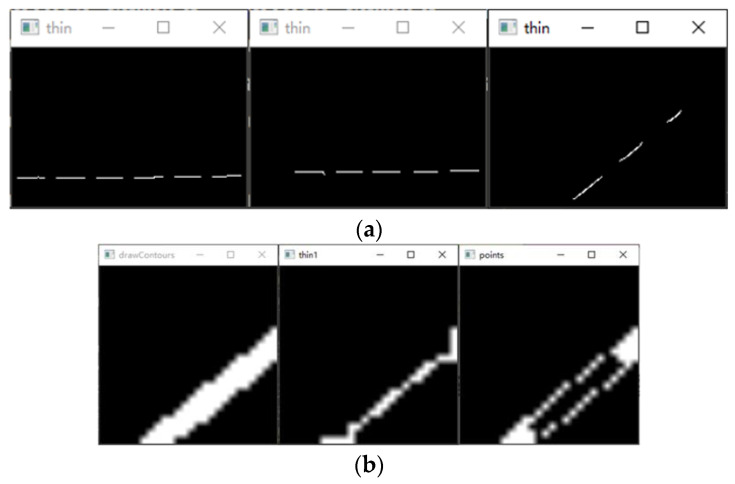
Structured light strip profile refinement results. (**a**) Preliminary refinement results; (**b**) Refined results after slicing.

**Figure 10 sensors-23-00848-f010:**
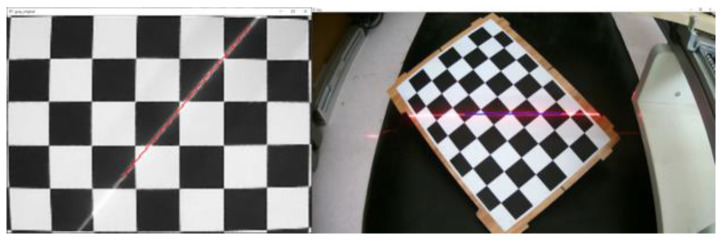
Structured light strip centerline extraction results.

**Figure 11 sensors-23-00848-f011:**
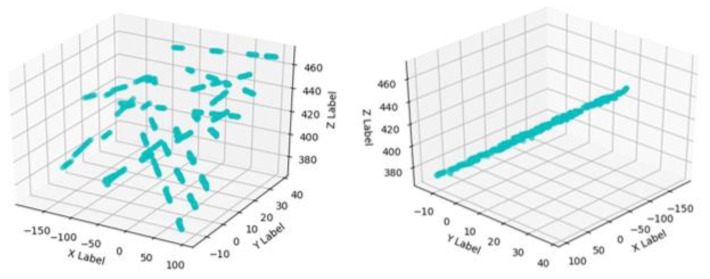
Calculated camera coordinates for each set of structured light strip quadrature points.

**Figure 12 sensors-23-00848-f012:**
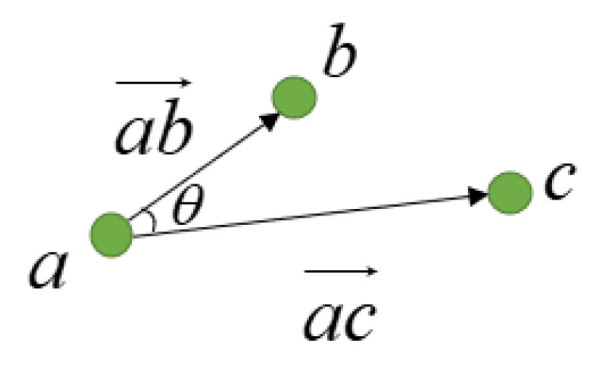
Linear constraint diagram.

**Figure 13 sensors-23-00848-f013:**
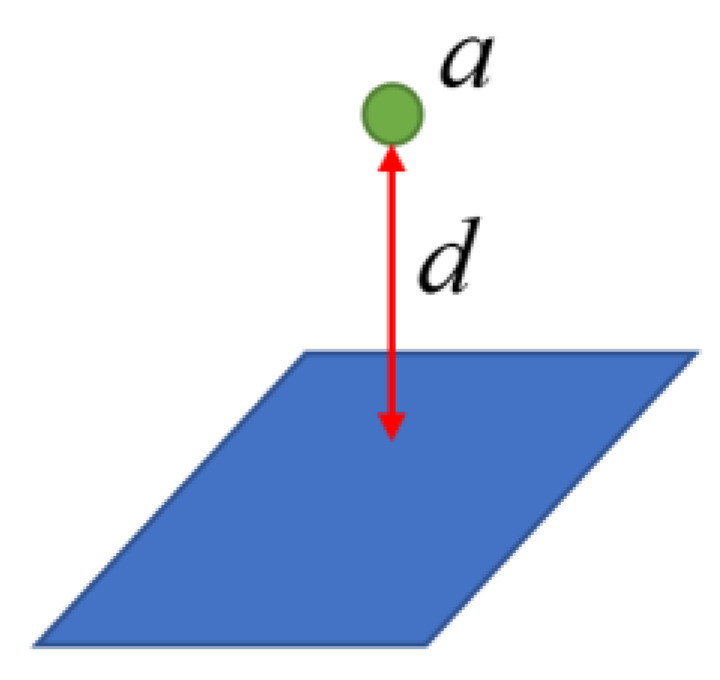
Planar constraint diagram.

**Figure 14 sensors-23-00848-f014:**
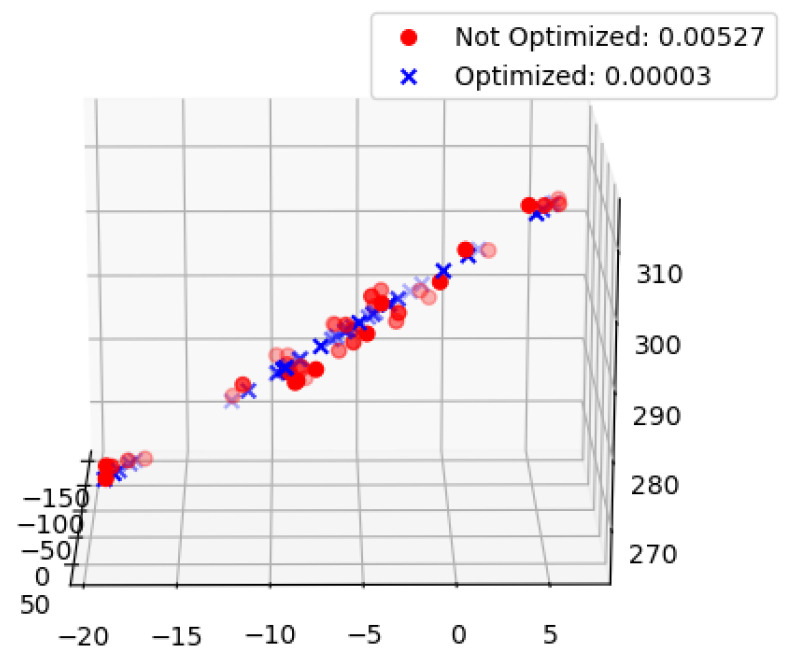
Single structure light plane optimization results.

**Figure 15 sensors-23-00848-f015:**
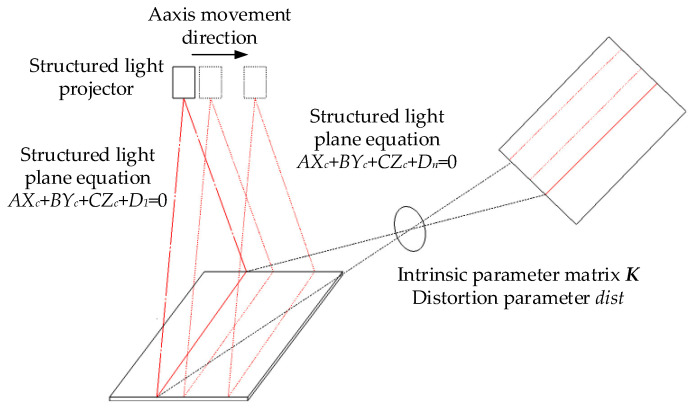
Diagram of calibration principle of the multi-structured light plane.

**Figure 16 sensors-23-00848-f016:**
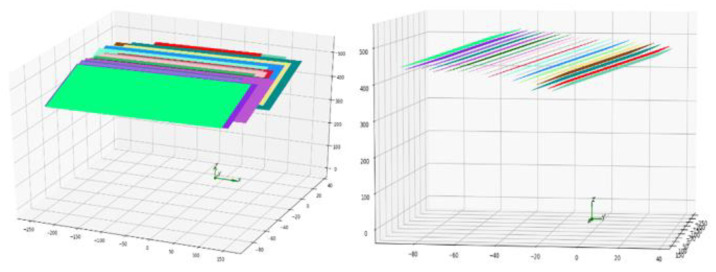
Multi-structured light plane optimization results.

**Figure 17 sensors-23-00848-f017:**
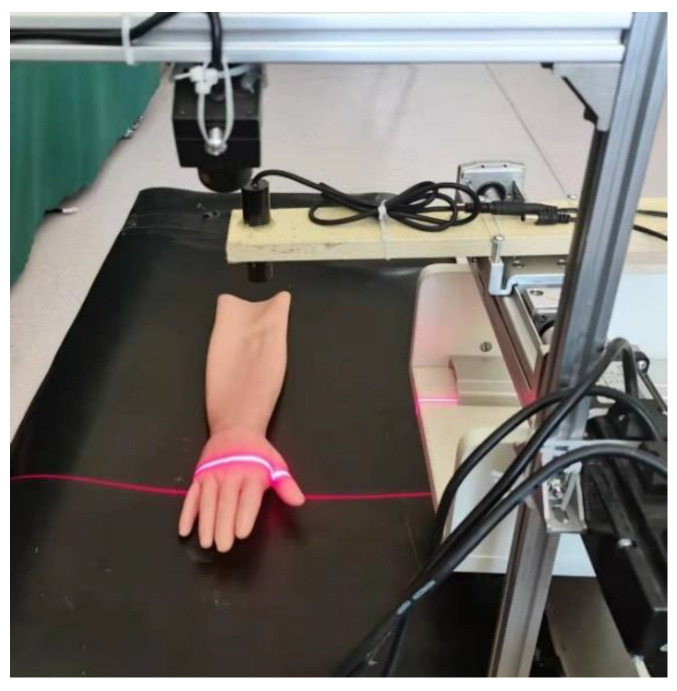
Experimental platform.

**Figure 18 sensors-23-00848-f018:**
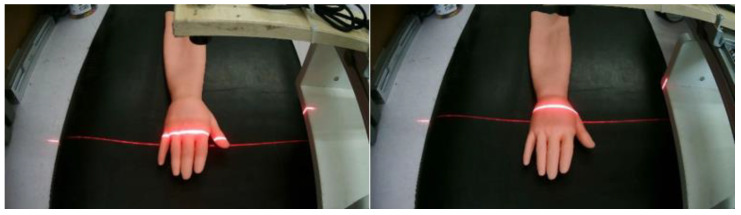
Acquisition of structured light scanning dorsal hand image.

**Figure 19 sensors-23-00848-f019:**
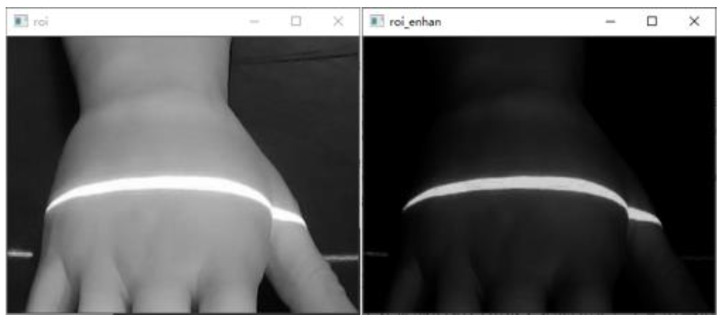
Grayscale image and gamma-transformed image of ROI area on the dorsal hand.

**Figure 20 sensors-23-00848-f020:**
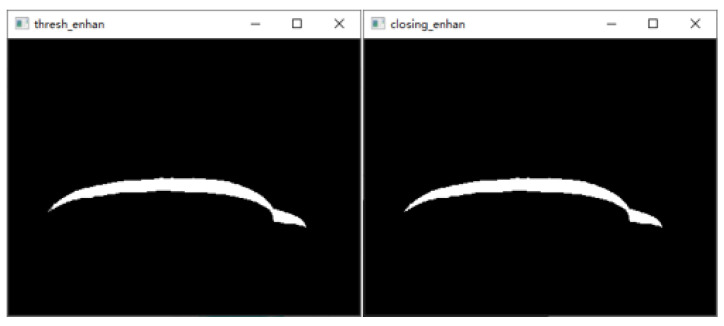
Binarization processing image and closed-operation processing image.

**Figure 21 sensors-23-00848-f021:**
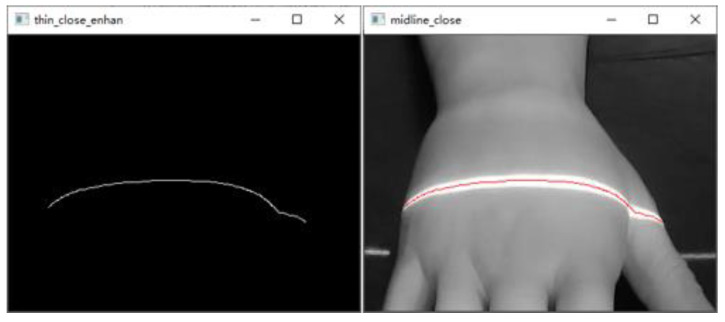
Refinement processing image with structured light strip extraction results.

**Figure 22 sensors-23-00848-f022:**
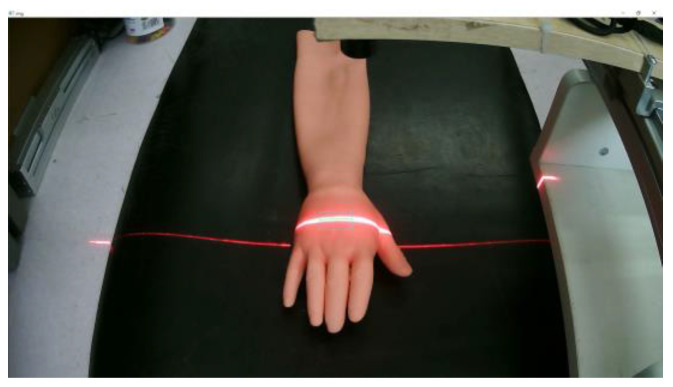
Structured light strip extraction results in the dorsal hand entry area.

**Figure 23 sensors-23-00848-f023:**
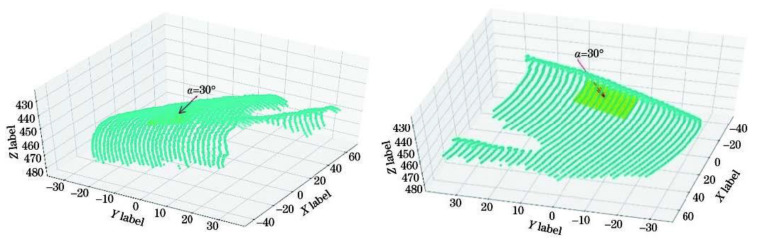
Plane fitting results for the dorsal hand into the needle area.

**Figure 24 sensors-23-00848-f024:**
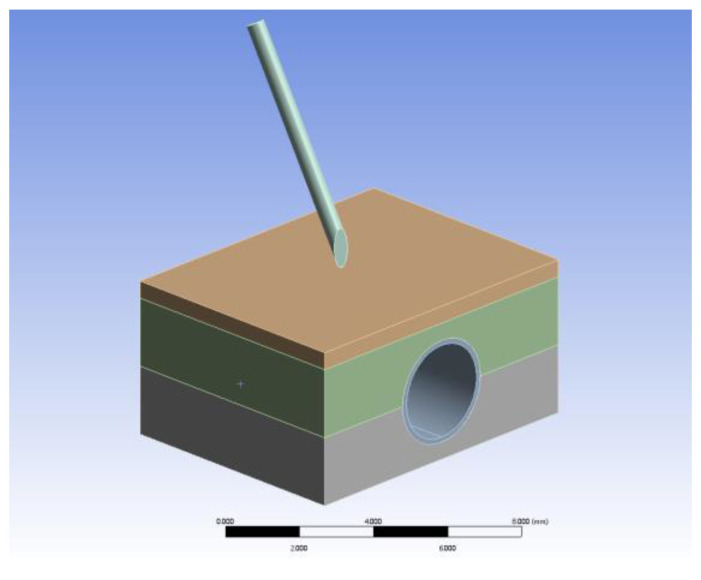
Dorsal hand venipuncture model.

**Figure 25 sensors-23-00848-f025:**
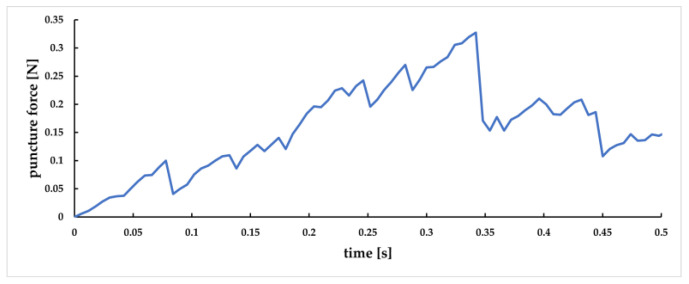
The variation curve of overall puncture force.

**Figure 26 sensors-23-00848-f026:**
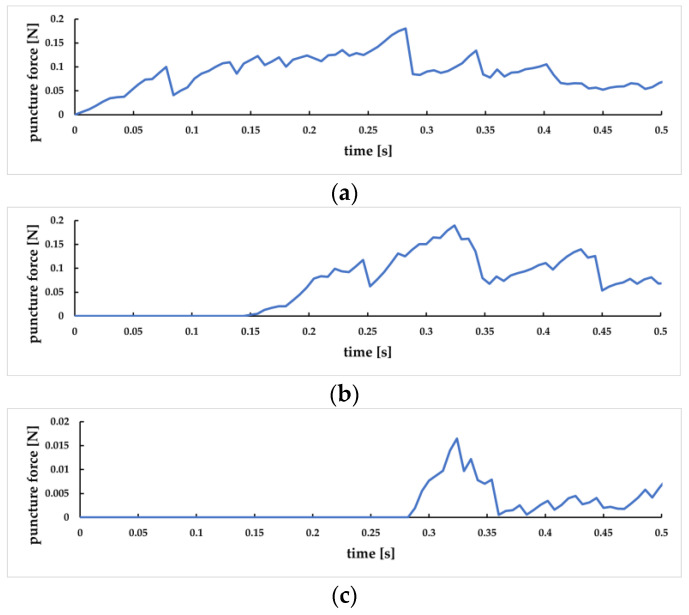
The variation curve of layered puncture force. (**a**) Change of skin layer puncture force; (**b**) Change of subcutaneous fat layer puncture force; (**c**) Change of venous vascular puncture force.

**Figure 27 sensors-23-00848-f027:**
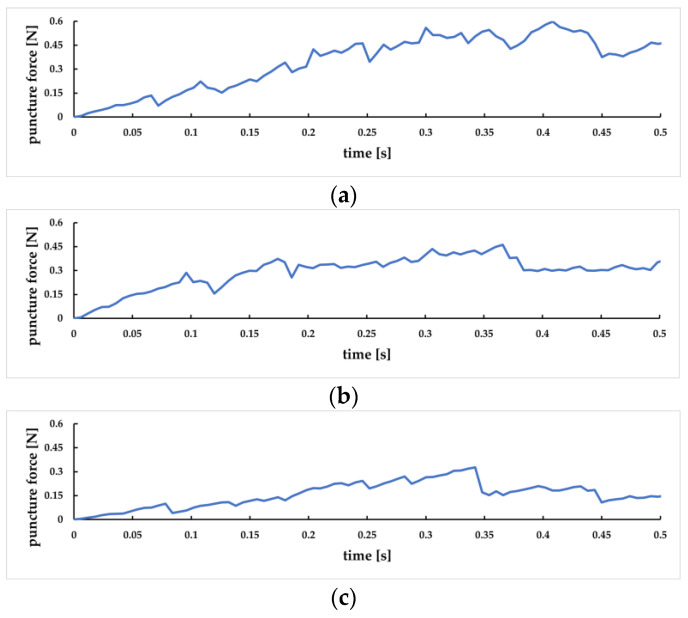
The variation curve of puncture force at different needle entry angles. (**a**) Change of puncture force at 30° needle insertion angle; (**b**) Change of puncture force at 45° needle insertion angle; (**c**) Change of puncture force at 60° needle insertion angle.

**Table 1 sensors-23-00848-t001:** Optimization results of reprojection error.

sNumber	Not Optimized	Optimized
1	56.171	3.064 × 10^−7^
2	38.708	3.204 × 10^−7^
3	134.205	8.626 × 10^−7^
4	45.723	2.098 × 10^−7^
5	41.746	7.590 × 10^−7^
6	45.070	2.935 × 10^−7^
7	45.809	8.534 × 10^−7^
8	59.161	3.765 × 10^−7^
9	43.016	3.023 × 10^−7^

**Table 2 sensors-23-00848-t002:** Multi-structured light plane fitting results.

Structured Light Projector Travel Distance	Distance between Two Planes *d*	Structured Light Projector Travel Distance	Distance between Two Planes *d*
5 mm	5.21	35 mm	4.83
10 mm	5.13	40 mm	5.08
15 mm	4.98	45 mm	4.84
20 mm	5.00	50 mm	4.91
25 mm	5.21	55 mm	4.98
30 mm	4.97	60 mm	5.03

**Table 3 sensors-23-00848-t003:** Structured light plane calibration results.

Structured Light Plane Common Normal Vector: (*A*, *B*, *C*)	Solving Parameters of Structured Light Plane Equation *D*: (*a*, *b*)
(0.007, −0.876, 0.482)	(−1.005, −191.717)

**Table 4 sensors-23-00848-t004:** Optimization results of structured light plane calibration.

Structured Light Projector Travel Distance	The Predicted Value of Structured Light Plane Equation Parameter *D*	The Real Value of Structured Light Plane Equation Parameter *D*	The Absolute Value of the Difference
0	−191.717	−191.428	0.289
5 mm	−196.743	−196.640	0.103
10 mm	−201.769	−201.71	0.002
15 mm	−206.795	−206.699	0.096
20 mm	−211.821	−211.701	0.120
25 mm	−216.847	−217.409	0.562
30 mm	−221.873	−222.175	0.302
35 mm	−226.899	−227.004	0.105
40 mm	−231.925	−232.084	0.159
45 mm	−236.951	−236.927	0.024
50 mm	−241.977	−241.839	0.138
55 mm	−247.003	−246.819	0.184
60 mm	−252.029	−251.849	0.180

**Table 5 sensors-23-00848-t005:** Table of mechanical characteristic parameters.

Organization	Materials	Mechanical Parameters				
		C10 (MPa)	C20 (MPa)	C01 (MPa)	D1 (MPa^−1^)	D2 (MPa^−1^)
Skin	Yeoh model	0.5	−0.06		0.01	0.1
Subcutaneous fat	Mooney Rivlin model	0.1		−0.04	0.01	
Venous vascular	Yeoh model	0.4	−0.05		0.01	0.12
Muscle	Mooney Rivlin model	1		−0.5	0.1	

## Data Availability

Data sharing not applicable.

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
