# Peer review of "Decision Method of Optimal Needle Insertion Angle for Dorsal Hand Intravenous Robot"

_sensors, 2023, doi:10.3390/s23020848_

Round 1

Reviewer 1 Report

The article concerns decision method of optimal needle insertion angle, which can be used for dorsal hand intravenous robot, decreasing the risk of e.g. COVID-19 infection.

It includes 4 sections containing an introduction, method description, results of the experimental verification and conclusion.

The subject of the article is interesting and current.

The strong point of the article is a novel and useful method.

Essential remarks:

1. The authors made an introduction to the subject of the paper, but it is not exhaustive enough. I propose to highlight the contribution of the publication more and to include more literature items related to the subject of the article.

2. Mathematical relationships need to be corrected. For example, equation (1) has a value of 0 and a vector of 0, which is indistinguishable, and it is not known what the dimension of vector 0 is. Also, the elements of the matrix in equation (1) are misaligned.

3. Please verify the consistency of the equation of the plane according to formulas (4) and (12) in Fig. 5, Fig. 15 and in line 322.

4. The pseudocode is badly written and is hard to understand. I suggest presenting it in accordance with generally applicable standards and describe it in more detail.

5. Please verify the symbol "x" in equation (6). It looks like a cross product, but it's probably just matrix multiplication.

6. The notation of equation (7) is incomprehensible. It is not known what the numbers 2 in the subscripts and superscripts mean when writing the norms "|| ||".

7. There are too many significant digits in equation (20). Please use adequate precision.

Editorial remarks:

1. I suggest using bold font for matrices and vectors for better readability.

2. Transformation matrix written as "t" looks strange, because it is more associated with time. I suggest using the more typical "T" designation.

3. I suggest changing the P_image and P_camera symbols, i.e. use subscripts instead of "_".

4. Space should be used everywhere when writing units, e.g. not "5mm" but "5 mm".

5. In equation (15) it should be "zn", not "z3".

6. Figures 25, 26 and 27 are difficult to read. Please significantly increase the font of the description of the axes and physical quantities and increase the thickness of the lines in the graphs.

Reviewer 2 Report

This paper discusses an optimal needle entry angle approaching direction for a dorsal hand intravenous injection robot. The operational environment is supported by the robot.

A vision sensing is relying on the approach that combines a 2D image acquisition and structured light that is projected to determine the z-depth. The error in localisation is acceptable.

The work is interesting and should be considered for the journal. However, some observations are as follows:

1)     the authors should reflect on using a developed technology within the real environment. For example, how would the system adapt to the micro-movements that a person may make during a procedure?

2)     Some sentences are simply too long and therefore hard to follow. For example:

“Qi Peng's team [10]at Tongji University proposed a method to determine the 47 entry angle of the vein in the elbow based on ultrasound assessment, which takes the angle between the needle axis and the whole arm of the patient as the entry angle and roughly selects a large or small entry angle by ultrasound assessment of the vessel thickness.”

or

Firstly, the image coordinate of the dorsal hand entry point is used as the center point to intercept the entry region, and the centerline of the structured light bar within the entry region is extracted separately and optimized using the above optimization method to obtain the camera coordinate points of the dorsal hand entry region.“

or

“According to the structured light plane calibration method proposed in this paper, the structured light plane equation at any position can be calculated, and then the dorsal hand can be scanned to extract the centerline of the structured light bar on each set of scanned images of the back of the hand, and the structured light plane equation at that position can be matched to obtain the spatial point clouds of the dorsal hand.”

or

“After the optimized dorsal surface of the needle entry area is obtained, the angle between the direction vector of the needle axis and the dorsal surface of the hand is the needle entry angle; simulation experiments are designed to investigate the changes in the penetration force under different needle entry angles, and the optimal needle entry angle decision is completed.”

or

“Through the previous work, our group has completed the recognition segmentation of dorsal hand vein images, extracted the dorsal hand vein pixel coordinates using the pruning algorithm-based needle insertion point location decision method (PT-Pruning), and obtained the optimal needle entry point and needle entry direction for dorsal hand vein by considering the vascular cross-sectional area and bending value of each vein entry area [19] in a comprehensive decision.”

There are a lot of similar examples. Please, break such sentences into two or more parts to be easier to follow.

3)     The literature review section should be formatted in accordance with the journal template. The references are missing the DOI part. The reference list should be in English.

4)     There are also some errors in style, for example, check the line number 323. Please, carefully check the overall manuscript

Round 2

Reviewer 1 Report

The authors in the revised version of the manuscript took into account all recommendations, therefore the article is suitable for publication.

There was a misunderstanding regarding one editorial comment, i.e. I suggested using bold font for matrices and vectors for better readability, for example T, R, K, M and vector 0 (3x1), and not using bold for scalar elements, like u, v, 1 etc. Therefore, it can still be corrected in the final version of the article and does not require a re-review.

Reviewer 2 Report

The authors have fulfilled all my comments and suggestions.

Author Response

Thank you very much for your recognition of this article.